# Text-Guided Customizable Image Synthesis and Manipulation

Zhiqiang Zhang [1] , Chen Fu [1], Wei Weng [2] and Jinjia Zhou [1,*]

1   Graduate School of Science and Engineering, Hosei University, Tokyo 184-8584, Japan
2   Institute of Liberal Arts and Science, Kanazawa University, Kanazawa 920-1192, Japan
*   Correspondence: zhou@hosei.ac.jp

**Abstract:** Due to the high flexibility and conformity to people's usage habits, text description has been widely used in image synthesis research recently and has achieved many encouraging results. However, the text can only determine the basic content of the generated image and cannot determine the specific shape of the synthesized object, which leads to poor practicability. More importantly, the current text-to-image synthesis research cannot use new text descriptions to further modify the synthesis result. To solve these problems, this paper proposes a text-guided customizable image synthesis and manipulation method. The proposed method synthesizes the corresponding image based on the text and contour information at first. It then modifies the synthesized content based on the new text to obtain a satisfactory result. The text and contour information in the proposed method determine the specific content and object shape of the desired composite image, respectively. Aside from that, the input text, contour, and subsequent new text for content modification can be manually input, which significantly improves the artificial controllability in the image synthesis process, making the entire method superior to other methods in flexibility and practicability. Experimental results on the Caltech-UCSD Birds-200-2011 (CUB) and Microsoft Common Objects in Context (MS COCO) datasets demonstrate our proposed method's feasibility and versatility.

**Keywords:** artificially controllable image synthesis; image manipulation; generative adversarial networks

## 1. Introduction

Painting is a visual art but also a man-made art. With the rise of artificial intelligence, scientists have been committed to making machines intelligent bodies like humans and achieving many human behaviors, such as face recognition, text generation, and painting. One of the most difficult behaviors to achieve is image synthesis. Unlike straightforward information, such as category labels and text, images contain a wealth of information, making them more difficult for machines to understand. Therefore, image synthesis is a very challenging task for machines.

In recent years, machines have been performing better and better in image synthesis tasks due to the emergence of deep learning. Some outstanding models, such as variational autoencoders [1] and autoregressive models [2], can already generate attractive image synthesis results. Recently, the emergence of generative adversarial networks (GANs) [3] is a milestone in image synthesis. Inspired by the GAN, many works have been able to synthesize highly realistic image results. However, the original input of a GAN only contains noise vectors derived from Gaussian distribution or uniform distribution, making the synthesis process of the GAN-based image synthesis model artificially uncontrollable, leading to its poor practicability.

To improve the practicability of the GAN model, the conditional generative adversarial network (CGAN) [4] was proposed. A CGAN can realize the control of the composite image category and related attributes by introducing additional category labels and attribute information. For example, given a category label "bird", the model can synthesize an image with birds. However, it is difficult for this method to realize complicated image synthesis

due to the small amount of information contained. On the other hand, category labels or attributes do not conform to human input habits, resulting in the corresponding model not having good applicability.

To address this issue of CGANs, text-to-image synthesis (T2I) was proposed. A text description contains rich information such as content and colors and is intuitive to humans. Therefore, using text descriptions to generate images has recently attracted the attention of many researchers. Many research methods have been proposed in this field, and they have achieved stunning image synthesis effects. Although the research on text-based image synthesis has resulted in impressive achievements, there still remain several problems. Concretely, the text information cannot determine the specific shape of the synthesized image, resulting in some inferior results that may be included in the synthesized images (as shown in Figure 1a). On the other hand, T2I methods can only generate the corresponding image based on the input text, whereas they cannot use the new text description to modify the synthesized image and obtain satisfactory results. These problems lower the practicability of current T2I works.

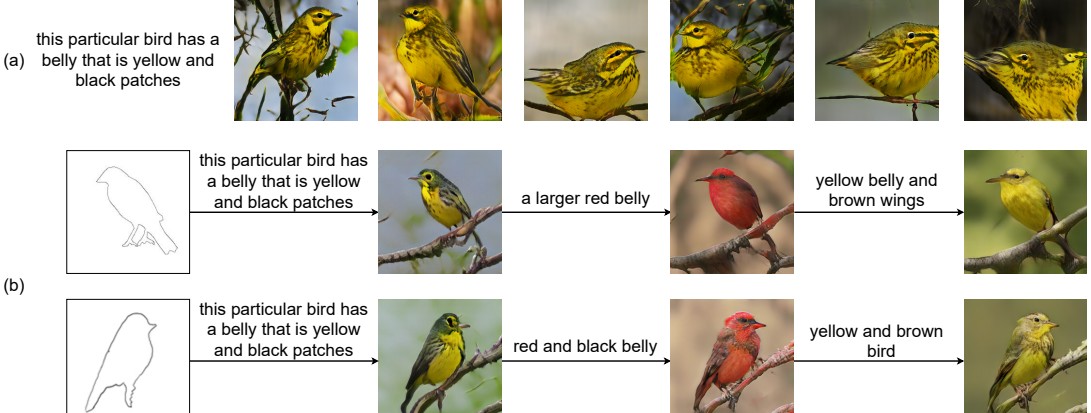

**Figure 1.** (**a**) The synthesized results of AttnGAN [5]. Given the input text description, AttnGAN synthesizes multiple image results with different qualities. (**b**) The upper contour was obtained by processing the dataset image, and the lower contour was obtained by drawing manually. Based on text and contour information, our method can synthesize the corresponding image. Then, the synthesized image content can be continuously modified based on the new text.

To cope with these issues, we propose a text-guided customizable image synthesis and manipulation method with two modules: (1) a customizable image synthesis module, which leverages the text description and contour information to synthesize the corresponding image result, and (2) an image manipulation module, in which new text can be used to modify the previously synthesized image content to achieve a satisfactory result. The text description and contour information in the two modules allow manual input and conform to people's input habits. Thus, our method has high flexibility and practicability.

The main contributions of our work are as follows:

- We propose an effective and novel method that can perform customized image synthesis based on text and contour information and then utilize new text to modify the customized image content.
- Compared with recent T2I methods, our proposed method can basically ensure the synthesized image's authenticity because the contour information determines the basic shape. On the other hand, since the new text is allowed to modify the content of the previously synthesized image continuously (see Figure 1b), the whole work is more practical.
- The text description and contour information are interactive and intuitive to users. Therefore, our proposed method has excellent flexibility.

- The feasibility and versatility of our proposed method are verified on the two widely used datasets (CUB [6] and MS-COCO [7]).

## 2. Related Work

**Image Synthesis.** Currently, deep learning has made many breakthroughs in the image synthesis field. A variational autoencoder (VAE) [1] simulates the image synthesis as a probability graph model and uses the lower bound of maximizing the data likelihood to achieve the image synthesis task. Aside from that, generative adversarial networks (GANs) show better performance in image synthesis, and various GAN structures [8,9] perform arresting of image synthesis quality. Meanwhile, the conditional GAN [4] was proposed, and it can enable GANs to synthesize image results under various conditional information (such as category labels and text), which makes GANs highly scalable. The most critical problem with current label-based and text-based image synthesis research [5,10,11] is that the specific shape of the object to be synthesized cannot be determined, making the quality of the synthesis result unable to be guaranteed. As shown in Figure 1a, AttnGAN [5] can synthesize multiple image results based on the input text, but these results have different shapes and some have very poor quality. The main reason for the poor quality is that the shape of the object is not real. This situation lowers the practicability of current image synthesis methods.

**Customizable Image Synthesis.** In order to make image synthesis methods more practical, research on customizable image synthesis has gradually emerged in recent years. The layout- and contour-based image synthesis methods [12–14] can input layout or sketchy contour information to determine the shape information of the synthesized object, thereby achieving controllable image synthesis. These works can control the shape of the generated object well but cannot control the detailed content, lowering the practicality of this method. For example, inputting the contour of the bird can determine the shape information of the composite bird image but cannot control the color, texture, or other attributes of the bird.

Another customizable image synthesis method is based on text and additional information. The text can be used to determine the basic content, and the additional information can be used to specify the basic location or shape of the composite object. A GAWWN [15] achieves a certain degree of controllable synthesis by combining the text description and additional annotations. The additional annotations are used to control the location information of the composite object. Nevertheless, the additional annotation information used by a GAWWN has limited control effects. On the other hand, the quality of the GAWWN synthesis results is mediocre. In contrast, the text and contour all conform to people's input habits, and it can synthesize images based on text and contour information so that it has better artificial controllability and practicability. CustomizableGAN [16] achieves a customized image synthesis effect based on the text and contour, which is similar to our work. However, CustomizableGAN can only synthesize $128 \times 128$ images, and the overall quality is ordinary. In contrast, our work can synthesize image results with a higher resolution ($256 \times 256$) and better synthesis quality. Moreover, our work can continue to use new text to modify the content of the composite image, which makes our work more practical.

**Image Manipulation.** Image manipulation includes many research fields, such as image editing [17], image style transfer [18], and so on. Among those, the most relevant research to our work is image manipulation based on the text. An SISGAN [19] uses text information to modify the content of the original image for the first time and achieves a certain degree of a content modification effect. The modified image not only conforms to the text's semantic information but also keeps the basic content of the original image (such as the shape of the object) unchanged. Still, the overall modification effect of an SISGAN is mediocre. Later, the TAGAN [20] and ManiGAN [21] achieved a better image content modification effect based on the text description. Their modified images conform well with the input text's semantic information and have made significant progress in terms of authenticity. However, this method is usually used to modify the real image, which

limits its practicality. In this paper, we combine text-based customizable image synthesis and image manipulation to further improve the practicality. Specifically, we use the text and contour to achieve customizable image synthesis and then use the text-based image manipulation method to continuously modify the synthesized image content with new text to achieve satisfactory results. The whole process is artificially controllable, showing that our method has excellent practicability.

## 3. Our Proposed Method

### 3.1. Preliminary

The adversarial learning characteristics of the generative adversarial networks (GANs) make it have excellent scalability, and it can promote the research process for many fields [8,22,23], especially in image synthesis. GANs include a generator ($G$) and a discriminator ($D$), in which the design of the generator and discriminator structures are determined according to the specific task. In image synthesis, $G$'s task is to generate the fake image result and make $D$ believe it is true, and the task of $D$ is to distinguish the authenticity of the received image. The corresponding specific equation is as follows:

$$\min_G \max_D V(D,G) = \sum_{x \sim p_{data}}[\log D(x)]$$
$$+\sum_{z \sim p_z}[\log(1 - D(G(z)))] \tag{1}$$

where $x$ and $z$ represent the image in the dataset and the noise vector, respectively, while $p_{data}$ and $p_z$ represent the distributions corresponding to $x$ and $z$, respectively.

The specific equation for using conditional variables is as follows:

$$\min_G \max_D V(D,G) = \sum_{x \sim p_{data}}[\log D(x,c)]$$
$$+\sum_{z \sim p_z}[\log(1 - D(G(z,c)))] \tag{2}$$

where $c$ denotes the conditional variable, which can be a category label, text, contour, or other information.

### 3.2. Customizable and Manipulation Architecture

**Customizable Image Synthesis.** Figure 2 shows the customizable image synthesis structure. Initially, the text and contour information are encoded into the corresponding feature vectors by the bi-directional long short-term memory (Bi-LSTM) [24] and Inception-V3 [25] models. Then, these two features are combined and processed into the first-stage features through multiple upsampling operations. Afterward, the features are processed into the features of the next stage through the residual block [26] operation and upsampling operation. Simultaneously, the residual block and upsampling operations can be used to continue to obtain the next stage's features. In this paper, the device is limited to only the image features synthesized to the third stage.

For the contour features, in addition to synthesizing the image features, the affine transformation will be performed with the image features of each stage. As shown below in Figure 2, the contour features obtained at the beginning and the features of the first stage are affinely transformed to form the new features. Then, the new features are affine transformed with the next stage's features, and this process will be repeated until they are affine transformed with the last stage's features to form the final transformed features, which will be used for the final custom image synthesis. The whole affine transformation process is as follows:

$$h_i = \begin{cases} Up(concat(fea\_t, \ fea\_c)) & i = 1 \\ Up(resnet(h_{i-1})) & i > 1 \end{cases} \tag{3}$$

$$fea\_c^{new} = conv(fea\_c) \tag{4}$$

$$h_1^{new} = h_1 \cdot W(fea\_c^{new}) + b(fea\_c^{new}) \tag{5}$$

$$h_i^{new} = h_{i-1}^{new} \cdot W(conv(h_{i-1}^{new})) + b(conv(h_{i-1}^{new})) \tag{6}$$

where $fea\_t$ represents the text feature and $fea\_c$ represents the contour feature. $Up$, $concat$, $resnet$, and $conv$ stand for upsampling, concatenation, residual block processing, and convolution operations, respectively. $W$ and $b$ are the weights and biases learned using two convolution layers, respectively.

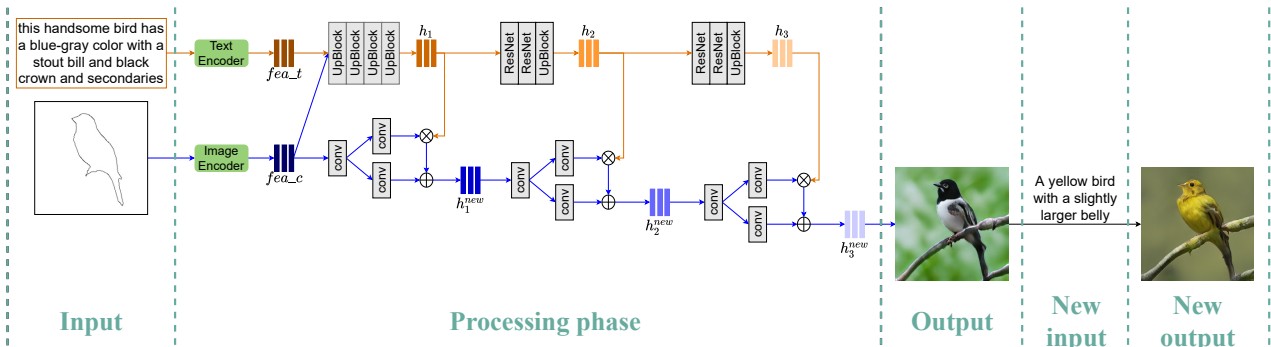

**Figure 2.** The basic structure of synthesizing images based on text and contour is shown in the figure above. Through continuous processing and fusion of text and contour information, the entire structure finally synthesizes the image result that not only conforms to the text's semantic information but also maintains the contour shape. The image manipulation based on the new text is shown on the far right. The image manipulation result shows that the new text's semantic information is well reflected in the previously synthesized image.

**Discriminator.** In the T2I research, the image and text matching-aware method [10] is widely used and has shown its effectiveness in enhancing the performance of image synthesis based on text information and improving the semantic consistency between an image and text. Therefore, we also adopted the matching-aware method for our work. Specifically, our discriminator receives three types of input: the generated image ($I_g$) + matched text ($T$), the original image ($I_r$) + matched text, and the original image + mismatched text ($T_{mis}$). The target of the discriminator is to recognize the first and third cases as false and the second case as true. In addition, it also needs to be able to correctly recognize true and false between the synthesized image and the original image.

**Image Manipulation.** The basic structure and training process of image manipulation is basically the same as that of customizable image synthesis. The difference is that the input in the customizable image synthesis structure is a combination of the text and the contour map, while the input in the image manipulation structure is a combination of the text and the original image. After the image manipulation model is trained, it will be used to modify the result of the customized image synthesis stage. As shown in Figure 2, based on the image manipulation model, new text can be continuously input to modify the content of the previously customized composite image.

### 3.3. Generator Implementation Details

In the generator, we use conditional augmentation, instance normalization, and a word-level attention mechanism, and their specific implementation details are as follows:

**Conditioning Augmentation.** For text processing, we use the conditioning augmentation ($CA$) [11] technology to generate more conditional variables to smooth the text representation. The specific CA equation is as follows:

$$D_{KL}(\mathcal{N}(\mu(\phi_t), \textstyle\sum(\phi_t)) \,||\, \mathcal{N}(0, I)) \tag{7}$$

where *KL* represents the Kullback–Leibler divergence, $\mathcal{N}$ denotes the Gaussian distribution, $\phi_t$ represents the encoded text vector, and $\mu$ and $\sum$ denote the mean and diagonal covariance matrix operations, respectively.

**Instance Normalization.** Since instance normalization (*IN*) [27] is more suitable for image style conversion tasks than batch normalization (*BN*) [28], the *IN* operation is used in the generator. The relevant calculation of *IN* is as follows:

$$\mu = \frac{1}{HW} \sum_{l=1}^{W} \sum_{m=1}^{H} x_{lm} \tag{8}$$

$$\sigma^2 = \frac{1}{HW} \sum_{l=1}^{W} \sum_{m=1}^{H} (x_{lm} - \mu)^2 \tag{9}$$

$$y = \frac{x - \mu}{\sqrt{\sigma^2 + \varepsilon}} \tag{10}$$

where $\mu$ is the mean, $\sigma$ is the covariance, $x$ and $y$ represent the input and output of *IN*, respectively, and $H$ and $W$ represent the height and width of the input vector.

**Word-Level Attention Mechanism.** Since the attention mechanism has achieved great success in text-to-image synthesis and has been widely used, it is also employed in our work. We chose the attention mechanism in [29] as the baseline. It utilizes the spatial and channel attention mechanisms to combine word features and each stage's image features more closely to generate higher-quality results. The specific equation of the attention mechanism applied to this work is as follows:

$$w_i = Bi\_LSTM(T), i = 1, 2, 3, \ldots, N \tag{11}$$

$$r_i = column(h_j), i = 1, 2, 3, \ldots, N_c \tag{12}$$

where $Bi\_LSTM$ represents the text encoder used, $w\_i$ represents the word features, $N$ represents the number of words, $h\_j$ represents the image features of the $j$th stage, $r\_i$ represents the $i$th image regional features, which corresponds to the column in the $h\_j$ vector matrix, $N\_c$ represents the total number of columns in $h\_j$, $\forall j \in \{1, 2, 3\}$, and $h_1$–$h_3$ represent the first stage's features up to the third stage's features. After the word and image region features are obtained, the similarity probability between them will be calculated:

$$c_{i,j} = \frac{exp(conv(w_i)^T * r_j)}{\sum_{k=1}^{N} exp(conv(w_k)^T * r_j)} \tag{13}$$

where $c_{i,j}$ represents the semantic similarity probability between the $i$th word features and the $j$th regional features, *conv* represents a convolution layer, which is used to process word features so that it can perform matrix calculations with image region features, and "$*$" denotes the Hadamard product. According to $c_{i,j}$, the output of spatial attention is as follows:

$$o_j = \sum_{i=1}^{N} c_{i,j} * conv(w_i) \tag{14}$$

The subsequent channel attention is found by first combining the results of spatial attention with word features to update the similarity probability and then using the updated similarity probability to calculate the new output:

$$c_{i,j}^{new} = \frac{exp(conv(w_i)^T * o_j)}{\sum_{k=1}^{N} exp(conv(w_k)^T * o_j)} \tag{15}$$

$$o_j^{new} = \sum_{i=1}^{N} c_{i,j}^{new} * conv(w_i) \tag{16}$$

Based on the baseline attention mechanism, we also explore the attention mechanism method combined with dynamic memory [30]. Different from the baseline, dynamic mem-

ory first calculates the importance of each word and then combines the word and regional features according to the importance of the words. Then, it calculates the similarity between the combined features and regional features:

$$g_i^w = \alpha(Tr(w_i) + Tr(\overline{R}))$$  (17)

$$w_i^c = conv(w_i) * g_i^w + conv(\overline{R}) * (1 - g_i^w)$$  (18)

where $\alpha$ denotes the sigmoid function, $\overline{R} = \frac{1}{N}\sum_{i=1}^{N} r_i$, $Tr$ represents matrix transformation, which can make $\overline{R}$ and $w_i$ perform matrix operations, and $w_i^c$ represents combined features, which will replace the word features in Equations (19) and (21) for consistency calculation.

Aside from that, the output in the baseline (such as in Equation (22)) will be combined with the previous regional features to realize updating of the regional features:

$$g_i^r = \sigma(W(o_i^{new}, r_i) + b)$$  (19)

$$r_i^{new} = o_i^{new} * g_i^r + r_i * (1 - g_i^r)$$  (20)

where $g_i^r$ represents the fusion information, and $W$ and $b$ represent the weight and bias, respectively.

### 3.4. Loss Function

**Generator Loss.** The loss function of the generator in the network structure includes two aspects: one is the GAN's adversarial loss, and the other is the perceptual loss. The specific adversarial loss of the generator is as follows:

$$L_{G\_adv} = \underbrace{-\frac{1}{2}\sum_{I_g \sim P_G} \log D(I_g)}_{unconditional\ loss} \ \underbrace{-\frac{1}{2}\sum_{I_g \sim P_G} \log D(I_g, T)}_{conditional\ loss}$$  (21)

The unconditional loss is used to determine whether the generated image is real, and the conditional loss is used to determine whether the synthesized image matches the text's semantic information.

The specific calculation of the perceptual loss is as follows:

$$L_{per}(I_g,\ I_r) = \frac{1}{CHW}\left\|\varphi(I_g) - \varphi(I_r)\right\|_2^2$$  (22)

where $C$, $H$, and $W$ represent the channel number, height, and width of the image, respectively, and $\varphi$ stands for the pretrained VGG model [31]. Following the work of Johnson et al. [32], we also used the second layer's feature output of VGG to calculate the perceptual loss.

In summary, the generator's loss function is as follows:

$$L = L_{G\_adv} + L_{per}$$  (23)

**Discriminator Loss.** The discriminator's loss function includes the adversarial loss, and the corresponding equation is as follows:

$$L_{D\_ture} = -\frac{1}{2}\sum_{I_r \sim P_{data}}[\log D(I_r) + \log D(I_r, T)]$$  (24)

$$L_{D\_false} = -\frac{1}{3}\{\sum_{I_g \sim P_G}[\log D(I_g) + \log D(I_g, T)] + \\ \sum_{I_r \sim P_{data}} \log D(I_r, T_{mis})\}$$  (25)

$$L_D = L_{D\_true} + L_{D\_false}$$  (26)

where $L_{D\_true}$ and $L_{D\_false}$ represent the cases where the discriminator needs to be judged as true and false, respectively. There are two cases that the discriminator needs to distinguish

as true: the original image ($I_r$) and the original image with matched text ($I_r, T$). There are three cases that need to be judged as false: the generated image ($I_g$), the generated image with matched text ($I_g, T$), and the original image with mismatched text ($I_r, T_{mis}$).

### 3.5. Training and Testing Details

During the training process, in the generator, Bi-LSTM [24] is used to extract the text feature, and Inception-V3 [25] is used to extract the contour feature. That aside, one convolutional layer and one InstanceNorm operation [27] are used in upsampling, and two convolutional layers and two InstanceNorm operations are used in the residual block. In the discriminator, four consecutive convolutional and BN [28] operations are used in the downsampling operation. Following the works of [5,16], we used an Adam optimizer [33] with an initial learning rate of 0.0002. Our model was trained in 600 and 120 epochs on the CUB and MS-COCO datasets, respectively. Due to device limitations, we set the batch size to 10. During the testing process, the entire structure and parameter settings were kept the same as during training.

## 4. Experimental Results

Our method was verified on the CUB [6] and MS-COCO [7] datasets. The basic information of these two datasets is shown in Table 1.

**Table 1.** The basic information for the CUB and MS COCO datasets.

| Dataset | CUB | | MS COCO | |
|---|---|---|---|---|
| | **Train** | **Test** | **Train** | **Test** |
| Number of images | 8855 | 2933 | 82,783 | 40,504 |

Since our work requires an input of contour information at the beginning, we first processed the images in the dataset to obtain the contour results. For the images in the CUB dataset, we directly obtained the contour results by processing the binary image provided by the dataset. For the images in the MS COCO dataset, we first extracted the foreground content according to the provided annotation file and then used the canny operator to process the foreground content and obtain the corresponding contour results. Some results after processing are shown in Figure 3.

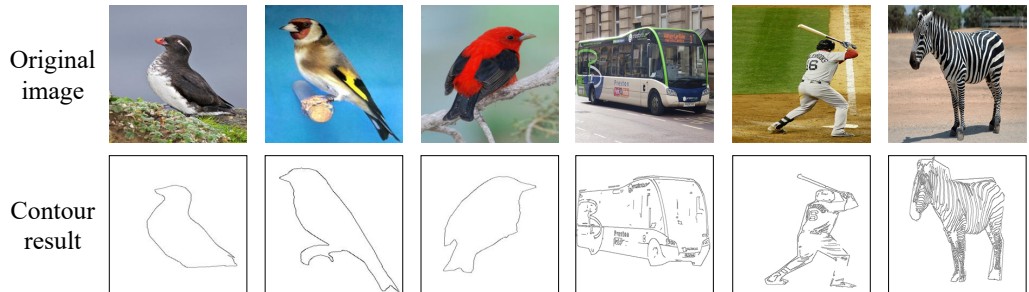

**Figure 3.** The processed contour results based on the original image are shown above.

### 4.1. Qualitative Results

**Customizable Image Synthesis Results.** The customizable image synthesis results on the CUB and MS-COCO datasets are shown in Figure 4. The input content included two parts: one was the text information used to determine the composite content, and the other was the contour map used to determine the composite object's shape information. The synthesized image results were not only in accord with the text's semantic information but also maintained the basic contour shape, which reflects the effectiveness of our method in customizing the synthetic image.

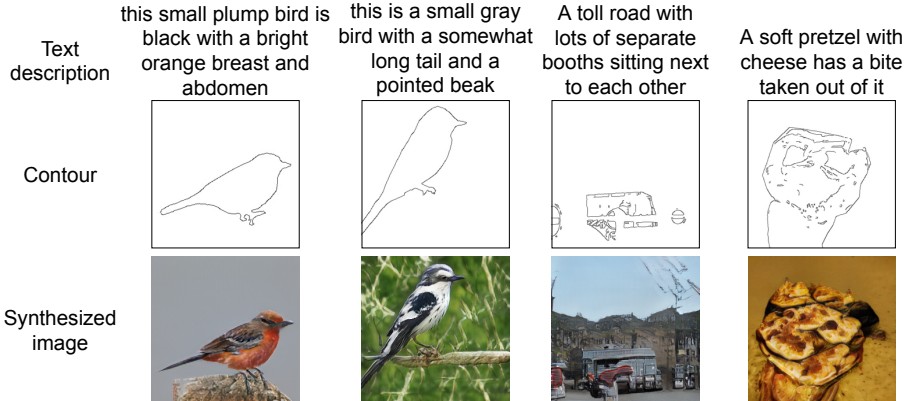

**Figure 4.** The customized synthesis results based on text and contour are shown above. The shape of the synthesized image is determined by the input contour, and the detailed content is determined by the text information.

**Text-Guided Image Synthesis and Manipulation.** The customized image synthesis and manipulation results based on text information are shown in Figure 5. Based on the input text and contour information, our method can generate the corresponding image results. It then allows continuous input of text information for modifying previously synthesized content. In comparison with directly inputting a long text description in the customization stage to synthesize the corresponding results, Figure 5 shows that our method can input simple text attributes to modify the previous synthesized image's content in the image manipulation stage, which shows the robustness of our method in manipulating the synthesized image with a few significant words.

**Comparison with T2I Methods.** The qualitative comparison results between our method and the current T2I methods are shown in Figure 6. From the bird synthesis results, the current T2I method, AttnGAN [5], could synthesize corresponding image results based on text descriptions. However, the quality of their synthesized results was mediocre. On the other hand, it can be clearly seen that the shapes of their synthesized results were different. Some shapes had poor authenticity, which made the synthetic results less authentic. In contrast, our results kept the contour's shape information, which guaranteed the authenticity of the generated results. Furthermore, they were consistent with the text's semantic information, which demonstrates the effectiveness of the proposed customized synthesis method under the text's guidance. The results of the complex image further reflect the superiority of our proposed method. Compared with the results synthesized by AttnGAN, the synthesized results of our method subjectively had excellent authenticity.

**Comparison with Custom Synthesis Methods.** The comparison results with other customized synthesis methods are shown in Figure 7. The first two lines show the input for customizable image synthesis, including the text description and contour. The next two lines reveal the results of CustomizableGAN [16] and our method. Overall, both methods could synthesize customized results based on text and contour information. However, the results of our method were better in terms of authenticity and clarity. In terms of details, such as eyes and textures, our results had superior performance.

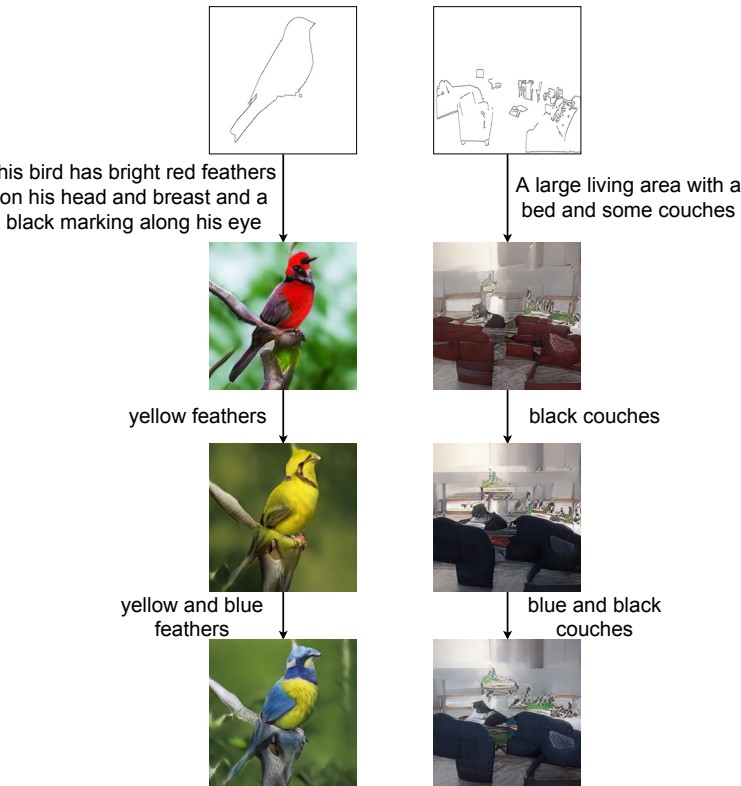

**Figure 5.** The above figure shows the results of image synthesis and manipulation based on text guidance. First, the image content was generated based on the text and contour information, and then the content of the synthesized image could be modified continuously by inputting the text of the object's attributes.

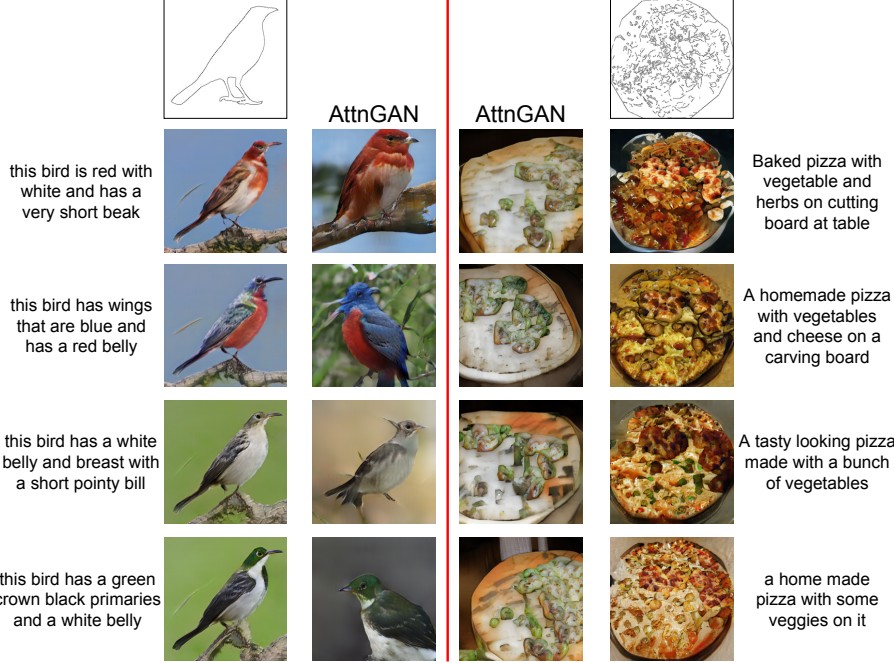

**Figure 6.** The qualitative comparison results between AttnGAN [5] and our customizable synthesis method on the CUB and MS COCO datasets are shown above.

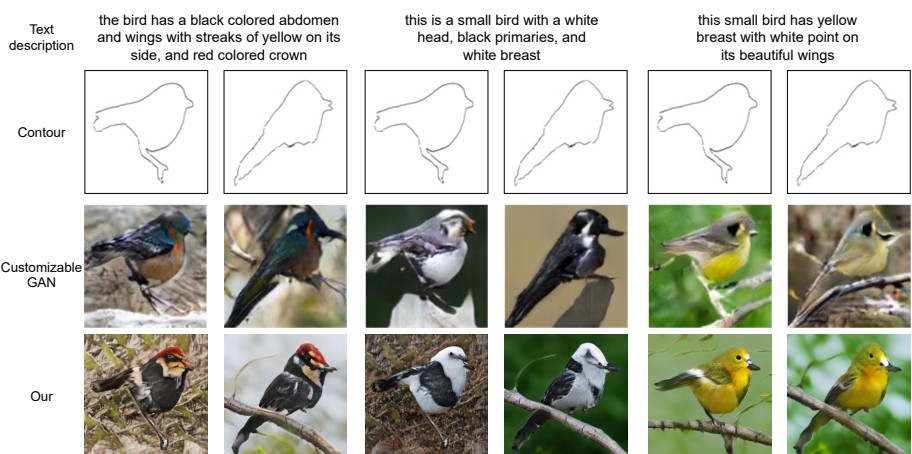

**Figure 7.** The qualitative comparison results between CustomizableGAN [16] and our customized method are shown above. Both the text and contour information in the figure were input manually.

*4.2. Quantitative Results*

We employed the Inception Score (IS) [9] and FID [34] as quantitative evaluation methods. The IS can be used to evaluate the diversity and authenticity of synthetic images, and FID can be used to evaluate the authenticity of synthetic images. Table 2 shows the quantitative comparison results.

**Table 2.** The quantitative comparison results among the T2I methods (a), custom methods (b), manipulation methods (c), and our method (d,e) are shown below.

| | | CUB | | MS COCO | |
|---|---|---|---|---|---|
| **Method** | **Model** | **IS ↑** | **FID ↓** | **IS ↑** | **FID ↓** |
| (a) T2I | GAN-CLS [10] | 2.88 | 68.79 | 7.88 | 60.62 |
| | StackGAN [11] | 3.7 | 35.11 | 8.45 | 74.05 |
| | StackGAN++ [35] | 4.04 | 18.02 | 8.3 | 81.59 |
| | AttnGAN [5] | 4.36 | 23.98 | 25.89 | 35.49 |
| (b) Custom | GAWWN [15] | 3.62 | 53.51 | n/a | n/a |
| | Customizable-GAN [16] | 3.12 | 65.36 | n/a | n/a |
| (c) Manipulation | SISGAN [19] | 2.24 | n/a | n/a | n/a |
| | TAGAN [20] | 3.32 | n/a | n/a | n/a |
| | ManiGAN [21] | 3.84 | 17.89 | 6.99 | n/a |
| Custom and (d) Manipulation (Baseline) | Custom stage | 3.98 | 23.19 | 15.03 | 46.02 |
| | Manipulation stage | 4.03 | 18.63 | 16.53 | 36.74 |
| (e) Baseline + DM | Custom stage | 4.02 | 18.72 | 15.27 | 45.04 |
| | Manipulation stage | 4.07 | 16.24 | 17.34 | 30.02 |

**Comparison with T2I Methods.** The last two rows (d,e) in Table 2 are our proposed method, including image custom synthesis and image manipulation. Compared with the T2I methods (Table 2a) that use only text for image synthesis, our method surpassed the performance of most methods. The FID performance was not weaker than that of the current excellent T2I methods. Moreover, since our method allowed manual input of the text and contour information as well as using new text to modify the composite image, it outperformed the T2I methods in terms of applicability.

**Comparison with Custom Methods.** Table 2b shows the results of current custom image synthesis methods. They use text combined with additional information (bounding box, key points, and contour) to synthesize images. In contrast, our customized method

outperformed them in terms of the IS and FID, demonstrating that our method had the best performance in customizable synthesis.

**Comparison with Manipulation Methods.** Table 2c shows the results of current image manipulation methods, in which text was used to modify the original image content. Our manipulation method showed superiority over them, too.

**Ablation Study.** Table 2d,e shows the ablation comparison experiment. We conducted ablation research on word-level attention, where baseline refers to the attention mechanism in [29] and Baseline + DM means that the dynamic memory method [30] was introduced on top of the baseline. The comparison demonstrates that the introduction of the dynamic memory method can improve the quality of synthesis results.

*4.3. More Discussion*

Our proposed method can achieve customizable image synthesis effects by inputting text and contour information, and it is able to manipulate the content of previously synthesized images by inputting new text information. Overall, our method has excellent practicality. However, there is still room for further improvement in our method in terms of overall practicability. Therefore, in order to further improve the practicality of the method, we suggest that further research be carried out in terms of the following aspects:

1. Enter new contours to modify the shape information of the generated image. For synthesized results using text and contours, our method currently can only continue to input new text to modify the content of the generated image but cannot input new contours to modify the shape information of the generated result. Therefore, to further improve the practicality of the method, we will explore the method that can use the new contour to change the shape of the image in the future.

2. Allow determining the background information of the image. Currently, our method can only control the foreground information of the synthesized image through the text and contour but cannot effectively control the background information. Controlling the background content can further improve the method's practicality, so this is worth researching in the future.

3. Allow changing the position and orientation of the composite object. Adjustment of the position and orientation is a common operation in image editing. Therefore, based on our method, achieving the operation of adjustment of the positions and orientations of synthetic objects can greatly improve the overall practicality.

4. Modify the texture content of the specific region. The modification effect of the existing method is to modify the overall object content, and it is hard to edit the regional texture content of the specific position of the object. Therefore, in order to achieve a more friendly interactive effect, it is necessary to research the corresponding method to achieve a more effective image content manipulation effect.

## 5. Conclusions

This paper proposes a customizable image synthesis and manipulation method based on text guidance. By using the text and contour information, the customized image result is synthesized first, and then the content of the previously synthesized image can be modified by continuing to input new text. Compared with the text-to-image synthesis method that cannot guarantee the authenticity of the synthesized image, our method ensures the basic authenticity of the synthesized image through the introduction of contour information so that it has better applicability. In comparison with the current customizable image synthesis methods and text-guided image manipulation methods, our method showed better performance, demonstrating our method's superiority. Aside from that, our method has the best practicality in synthesizing images, which promotes the development of image synthesis from the research field to the industrial application field.

**Author Contributions:** Conceptualization, Z.Z.; methodology, Z.Z.; writing—original draft, Z.Z.; investigation, Z.Z.; data curation, C.F.; validation, C.F.; writing—review and editing, W.W.; formal analysis, W.W.; project administration, J.Z.; supervision, J.Z. All authors have read and agreed to the published version of the manuscript.

**Funding:** This research was funded by the Joint Research Project of Young Research of Hosei University in 2021.

**Institutional Review Board Statement:** Not applicable.

**Informed Consent Statement:** Not applicable.

**Data Availability Statement:** Not applicable.

**Conflicts of Interest:** The authors declare no conflict of interest.

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
