# Peer review of "Text-Guided Customizable Image Synthesis and Manipulation"

_applsci, doi:10.3390/app122010645_

Round 1

Reviewer 1 Report

The paper presents a text guided customizable image Synthesis and manipulation method. The method first synthesis the image based in text and contour informations and then modifies the synthesised image based on new text description to obtain better results. 

The paper method description need modifications: figure block diagram before to figure 3 describes I/Ps, intermediate stages and O/Ps . Also algorithm is recommended to be added.

Section 3 and its subsection require to be backed up with figures of for examples: networks used, sample of images, texts, augmented texts, and more description of the GAN technique used,......etc.

New section should be added specific to the used CNN models with type of dataset that feeded for training, validation, and testing

Author Response

Thank you very much for your professional comments.

We have annotated the input, output, and intermediate processing phase in Figure 2.

Following your suggestion, we modified the order of the content, introduced the implementation details of the generator in Sec 3.3, and for the loss function in Sec 3.4, we also added more details (see lines 190-193).

In addition, for more training and testing details, we have added Section 3.5 to explain. At the same time, we added a description of how to obtain the contour results (see lines 208-214) and showed the processed contour results in Figure 3.

Reviewer 2 Report

Overall work is good and satisfactory

Justify the selection of the following parameters: Batch size, learning rate, epochs?

How did you fix different epochs on different datasets?

Add a few more results. 

In Fig. 4, the colour is changing but there isn't much change in texture. Is it possible to synthesise different texture or shape for the same object using text description? 

Is it possible to change the background of the object?

Is it possible to introduce a change in the position or orientation of an object through the textual description? e.g., rotate the pot on the table by 90 degrees in clock wise direction? etc.

Author Response

Thank you very much for your professional comments.

For the determination of batch size, learning rate, and epochs, we follow the existing related works (CustomizableGAN and AttnGAN). We illustrate this content in lines 200-202 of the current paper version.

For the changing of texture, shape, background, position, and orientation you mentioned, existing methods (including ours) can not do it efficiently. For this, we have specially added Sec 4.3 to the main content for more discussion, indicating where the follow-up research work can be carried out. The corresponding content is in lines 277-302.

In addition, we use Grammarly to check English writing and grammar to ensure that it is correct.